# Reinforcing and Reproducing Stereotypes? Ethical Considerations When Doing Research on Stereotypes and Stereotyped Reasoning

**Mathilde Cecchini**

Department of Political Science, Aarhus University, 8000 Aarhus C, Denmark; mcecchini@ps.au.dk

**Abstract:** Many social scientists are interested in studying stereotypes and stereotyped reasoning. This interest often comes from a wish to contribute to creating a more just and equal society. However, when we as scholars study stereotypes and stereotyped reasoning, we risk reproducing and maybe even reinforcing these processes, and thereby harming individuals or groups of individuals. The debates of this ethical issue mainly take the form of general discussions of research ethics and of weighing the aim of the research against potential harm to participants. While these reflections are extremely important, there is a need for discussing how this ethical issue can be handled in practice. The aim of this article is to develop a set of practical guidelines for managing this ethical issue, based on the examination of ethically delicate moments experienced during an ethnographic study of the construction of health and risk identities among seventh-graders in Denmark. Three guiding principles are proposed: Develop an ethical sensibility in order to identify ethically delicate moments; consider ethics as well as methods when constructing and posing questions; more specifically, briefings and debriefings can be used to address ethical issues; and, finally, make participants reflect upon their opinions and answers.

**Keywords:** research ethics; ethical sensibility; reflexivity; stereotypes; stereotyped reasoning; research with children; qualitative research; focus groups

## 1. Introduction

When we as scholars design and carry out studies, we are occupied with conducting efficient research in accordance with research criteria. We continuously encounter methodological hurdles, which we have to overcome to advance our research in the most efficient manner. However, sometimes we find ourselves in situations where conducting efficient research may conflict with ethical considerations or principles. Research projects examining stereotypes and stereotyped reasoning may very likely fall within this category. When scholars try to uncover how participants employ stereotypes in their categorization and identification of themselves and others, they risk reproducing and reinforcing stereotyped reasoning by drawing attention to or probing participants to employ specific categories or classifications when answering questions [1–3]. A researcher may be interested in uncovering class stereotypes and hence ask a respondent to assign specific characteristics to people from different social classes. Likewise, a scholar interested in uncovering gender stereotypes may ask participants to describe what they perceive as truly feminine and masculine. However, by doing so, the researcher invites respondents to engage in stereotyped reasoning. This constitutes a dilemma for the researcher: The concern about obtaining data and the ethical concern of not reinforcing processes of stereotyped reasoning and thereby potentially violating the principle of beneficence [4]. According to this principle, which is also referred to as the "do no harm" imperative of research, the potential benefits of research should outweigh the potential harms of participating for human subjects [4].

However, weighing potential harms and benefits is a task associated with a high degree of uncertainty since it is difficult to predict potential harms and benefits of research. Ethical review boards and codes of ethics are there to support and guide researchers in these ethical questions and to uphold the standard for ethical research. However, procedural ethics are not sufficient to address this ethical issue. Weighing potential harms and benefits requires significant knowledge about the research topic, the research setting and context as well as the methodological approach adopted in the concrete study, which ethical review boards may not always have, and which general ethical codes cannot take into account [5,6]. Moreover, since potential harms and benefits are difficult to predict, unexpected ethical issues can arise after approval by an ethical review board [5–7]. This calls for a focus on situated and contextualized ethics as well as ethics in practice. In other words, we have to discuss how we can minimize the risk of violating the principle of beneficence in practice in specific research situations.

The risk of reproducing stereotypes when doing research on stereotypes applies to a wide range of social science studies and is not unique to specific methodological approaches. However, few scholars have addressed this issue, and most discussions of the dilemma do not provide practical guidelines and substantial advice for scholars. The aim of this article is to develop a set of strategies that researchers can use, particularly in interview studies, to address the risk of reproducing or reinforcing stereotypes and stereotyped reasoning properly. The article is based on experiences from an ethnographic study of the construction of health and risk identities in the seventh grade in a Danish public school [8] as well as insights from the literature on these types of ethical concerns. Three guiding principles are derived from the examination of the empirical material as well as the engagement with the literature. First, scholars should seek to foster an ethical sensibility to be able to identify potential ethically delicate moments in the research process. Second, researchers should pay particular attention to the ethical dimension when they develop their questions. Third, they could ask participants to reflect upon their opinions and answers, for example, during interviews.

The article starts with a brief overview of how stereotypes and stereotyped reasoning have been studied in different literatures, what constitutes the ethical problem, and how scholars have sought to overcome it. Afterwards, the methodological approach and the study that forms the empirical base of the article are presented. The article then proceeds to the three guiding principles that could be helpful to scholars facing this ethical issue.

## 2. Studying Stereotypes and the Risk of Reproducing Stereotypes: The Dilemma in the Literature

Stereotypes and stereotyped reasoning are studied in various literatures based on a wide range of methodological approaches and methods. For example, experimental vignette studies are often used in the literature on stereotypes and discrimination in street-level bureaucracy [9–13]. Street-level bureaucrats such as caseworkers or teachers are presented with vignettes (case descriptions of a client) and asked to indicate how they would treat the client, for example whether they would impose sanctions, grant benefits, etc. By changing the name of the client (to an ethnic minority name) or some characteristics of the client (educational background, job), it becomes possible to study whether the client's ethnicity or social class affects how the street-level bureaucrat acts. This approach is often used in survey experiments, but can also be adopted in qualitative interviews [12,13].

Another approach to studying racial and social class stereotypes can be found in educational research in the form of statement questions in survey questionnaires. Respondents are presented with statements that represent prevailing stereotypes of race and class (for example: "Asians are better pupils than English pupils") and asked to agree or disagree. This approach has been criticized from an ethical point of view for encouraging and legitimizing the use of racial or ethnic stereotypes as frames of reference [3].

In gender studies, gender stereotypes are examined by, for example, asking participants what they associate with proper femininity and masculinity [14] as well as using the gender essentialism scale [15]. Within psychological research, a common way to study stereotypes (gender, racial, etc.) is

through the implicit association test, which measures the strength of association between concepts (for example women and men) and evaluations (clever and caring) [16,17].

Which methodological approach to adopt depends on the subject of study and the scope of the research, for example whether the researcher is interested in implicit biases (for example the implicit bias approach) or more conscious stereotypes and opinions (as in the case of statement questions). However, I would argue that all approaches entail a risk of reproducing or reinforcing stereotypes and stereotyped reasoning. Research on stereotypes and stereotyped reasoning shows how creating an in-group and an out-group in itself results in discrimination of the out-group [18,19]. This is what makes research on stereotypes and categorization important and relevant. However, this should also make researchers aware that drawing attention to distinctions between groups or categories of people in research situations may have consequences.

While the risk of reproducing and reinforcing processes of stereotyped reasoning is inherent to studying these phenomena, few scholars actually address and problematize the ethical implications of this when presenting their research. This ethical issue is mainly debated on more abstract levels in general discussions of research ethics that center on the importance of reflecting on these issues, on considering the purpose of our research and "whose side we are on" and on weighing the aim and scope of the research against potential harm [1,20]. While these contributions are extremely important, I would argue that researchers would benefit from a stronger linkage between these ethical considerations and their everyday research practice.

Some scholars have discussed how to accommodate the ethical dilemma on a more practical level. In her study on identity formation among ethnic minority children in the Danish public school, Gilliam discusses how she tried, during interviews with the children, to uncover the use of ethnic categories among the children and how this process increased the children's focus on ethnicity [2]. This may have reinforced specific identities and stereotype understandings among the children. Gilliam describes how she tried to question the boundaries the children drew during the interview in order to counter this side effect of the research process [2], but she does not provide evidence for whether this strategy was effective. Other scholars have argued that asking open-ended questions instead of statement questions, such as "Asians are better pupils than English pupils" and inviting respondents to agree or disagree, is a way to avoid reproducing and legitimizing specific stereotypes [3]. However, whether this is a more ethically appropriate strategy can be questioned, since asking statement questions may actually force respondents to reflect explicitly on the stereotype presented, whereas open-ended questions may allow respondents to answer based on implicit and unquestioned stereotypes. Which approach to choose is thus more a question of methodology than of ethics since changing the way of asking questions may also alter what the researcher is actually capturing or measuring.

The aim of this article is to advance and specify these insights from the literature by examining ethically delicate moments in a research project I recently conducted and to derive some practical strategies to accommodate the ethical concern of reproducing and reinforcing stereotypes and stereotyped reasoning. In the following section, I present the methodological approach adopted in this article as well as the study that forms the empirical basis of the article.

## 3. Materials and Methods

The aim of this article is to provide a set of guiding principles for how to address the risk of reproducing and reinforcing stereotypes and stereotyped reasoning in research on these issues. As mentioned, the article builds on discussions in the existing literature and experiences from an ethnographic study on the construction and enactment of health and risk identities in the seventh grade in Danish a public school [8].

I develop three guidelines that scholars should pay attention to and incorporate in their research in order to manage the risk of reproducing and reinforcing stereotypes. The guiding principles are derived by drawing on discussions in the literature and by examining ethically delicate moments in the empirical material. My approach can thus be characterized as abductive; I have moved back and

forth between theory and empirical observations, building claims through this iterative process [21,22]. In the following, I present the research design and methodology of the study.

The study took the form of an interpretive ethnography [23,24] on the topic of health risk prevention in schools conducted in four seventh-grade classes at two Danish public schools. More specifically, the project focused on how the meaning of health and risk as well as health and risk identities were constructed and transformed in the interaction between students, their peers and their teachers. The study asked the following questions: How are health and health risks defined in policies, among teachers and among students? How do policies, teachers and students categorize healthy schoolchildren and schoolchildren at health risk? How are identities as healthy and at risk performed in the school setting? The study thus examines categorization and classification processes as well as potential processes of stigmatization of individual students and groups of students. Since health is a potentially very sensitive topic related to stigma, this was a probable risk. Moreover, since the research was situated in the natural environment of the students (peers from school), the potential reinforcing effect of stigmatization has direct consequences for them (something they may experience in their daily lives at school).

The data was generated through a combination of participant observation (more than 500 h), focus groups with students, semi-structured interviews with teachers, focus group interviews with teachers and collection of policy documents. In this article, I mainly draw on experiences from the focus groups with the schoolchildren and observations. The observational data consists of observations of interactions between students and between teachers and students that somehow concerned situations where health risk behavior or state were at play, for example interactions where students discussed eating habits, health education, physical education classes, etc. The observational data consists of field notes on participant observation [24,25]. This entailed taking short notes in a notebook—a condensed description—during fieldwork and later re-writing the notes and filling in the gaps with details, thereby turning them into an expanded account [25].

I chose to conduct focus group interviews with the students because this allowed me to observe how the students negotiated the meaning of health and risk, as well as how they constructed health identities in interaction with each other [26,27]. Hence, focus groups were an appropriate technique to generate the type of data I needed to shed light on my subject of study. Furthermore, I was dealing with schoolchildren—a group of participants many researchers consider vulnerable. Research participants may have different needs and interests as well as varying degrees of power to pursue these interests and protect themselves, and accordingly some participants are in some situations vulnerable [1]. Schoolchildren may have less power to pursue their interests and protect themselves. Moreover, the relationship between adults and children entails an imbalanced power and information structure, which is not solely a result of the interview and research situation, but a general condition resulting from the fact that adults appear as authorities in every aspect of the child's life [28–32]. While it can be argued that the literature sometimes neglects the situational and dynamic nature of power relations among individuals and accordingly overestimates the imbalanced power structure between the adult researcher and children, doing research with children inevitably involves some methodological challenges. It is likely that it is more difficult to construct questions in a manner that makes them immediately understandable to the respondents, depending on their age. It may also make it more complicated to get answers since children may mistake the research situation for a teaching situation and wish to please the researcher by giving the right answers, or what they believe the researcher thinks are the right answers [2]. The imbalanced power structure cannot be eliminated, but it can be minimized by adopting "varied and imaginative research methods" [32]. One way is to make use of focus groups, which resemble a situation that schoolchildren are familiar with, namely interacting and talking with their peers. The focus group may have a more informal atmosphere and soften the asymmetrical power structure between the (adult) researcher and the (child) respondent [2]. Schoolchildren may feel more in control of the situation than in a single-person

interview with a researcher. This way, respondents are empowered, which is essential in order to reduce the power differential between adult researcher and child respondent [28].

Since I was interested in collective negotiation and construction of meaning and identities, I made sure to address the children as a collective and encouraged them to discuss the questions collectively. A way to foster discussions and negotiations in a focus is to make use of exercises [33]. Exercises give members of a focus group a common task to solve, a common point of departure for discussion, and it allows the researcher to facilitate rather than lead the research situation. During the focus groups, I thus made use of various exercises. First, before the interview, I asked the pupils to make a short photo diary from their everyday life with pictures of situations, activities, habits, etc. that they associate with being healthy and unhealthy. This exercise was inspired by the technique photovoice, which is used in community-based participatory research [34–37]. The pupils sent their photos to me via email or text, and I printed them and brought them to the focus group. These photo diaries then formed the basis of discussions in the focus group. Among other things, I asked the children to classify the photos from the photo diaries and to categorize and classify themselves and their peers in relation to friendship groups and health behavior and state. The classification exercises combined with photo material proved a good tool for this specific kind of research, because having photo material and exercises made it easier for the young participants to express their views on an abstract and intangible phenomenon such as health. Moreover, it made the focus group situation interesting and increased the participants' attention span. Finally, the fact that the schoolchildren took and brought their own photos seemed to empower them by letting them express their understandings and opinions without the researcher having to ask a lot of questions and taking the lead in the focus group. The interview guide is attached in full length in the Appendix A.

I had many methodological considerations and arguments for choosing this approach. However, studying health categorization through this specific approach and dealing with vulnerable participants (young teenagers) in their natural environment created an ethically delicate situation. In Denmark, there is no tradition for having ethical review boards that approve research projects in the humanities and social sciences. Research projects with human subjects in the natural and medical sciences do need to be approved by the national research ethics committee, which is an independent authority under the ministry of health [38]. Recently, some Danish universities have begun to establish local institutional review boards [39]. The purpose of these institutional review boards is to review research projects with human participants in the social sciences and humanities in cases where the researchers need the approval for example in order to publish in particular journals or obtain a grant. At the point where I conducted my research these institutional review boards had not yet been established, therefore my research proposal could not be approved by such an authority. Instead, I discussed potential ethical challenges and how to address them with colleagues, and I obtained informed consent from the school principals, teachers and parents. However, I still encountered ethical challenges during my research. Based on these experiences and reflections on discussions in the literature, the following section seeks to develop a set of guidelines for managing these kinds of ethically delicate moments.

## 4. Addressing the Dilemma

Studying stereotypes and stereotyped reasoning will most often involve drawing participants' attention to stereotypes or inviting participants to engage in stereotyped reasoning. Moreover, when researchers ask participants such questions, they may perceive it as a legitimization of the stereotypes in question. This last point may be more pronounced in qualitative research, such as interview studies, where the researcher's courtesy and responsiveness could be interpreted as a declaration of agreement. Studying categorization and stereotyped reasoning will thus inherently involve some risk of reinforcing or reproducing those processes. This is the case for this type of research independently of research design and specific methods used. This article focusses on how to address the dilemma in qualitative research, particularly ethnographic studies and interview studies, but I would argue that the guidelines could be useful for scholars working with, for example, survey questionnaires.

Even though research on human subjects will always influence the participants and the social world surrounding them to some extent—and potentially inflict some degree of harm to individuals—we should not, of course, refrain from doing research. Shedding light on stereotypes and stereotyped reasoning is highly relevant and important for society, but we should not neglect these ethical issues related to our research. Even though I do not believe it is possible to eliminate the risk of reinforcing these processes, I will argue that there are ways of minimizing it. The practical guidelines for minimizing this risk proposed below are by no means a substitute for procedural ethics such as general ethical codes and ethical review boards. They are suggestions for how researchers can manage ethically delicate moments in practice, which is rarely a topic in the literature.

*4.1. Cultivating an Ethical Sensibility*

Scholars have argued that ethical research behavior requires more than ethical knowledge and cognitive choices, namely that researchers are able to identify ethical issues and feel a responsibility to act in a morally appropriate manner [40]. Reflexivity has been stressed as a way to ensure ethical research by several scholars [7,41–43]. It is argued that researchers should not just be reflexive regarding the process of knowledge production for example concerning their positionality (methodological issues), but also in relation to how their research might affect research participants and how they as researchers should act in potential delicate situations (ethical issues) [7]. This kind of ethical reflexivity entails:

> "[A]n acknowledgment of microethics, that is, of the ethical dimensions of ordinary, everyday research practice; second, sensitivity to what we call the "ethically important moments" in research practice, in all their particularities; and third, having or being able to develop a means of addressing and responding to ethical concerns if and when they arise in the research (which might well include a way of preempting potential ethical problems before they take hold)" [7].

This section seeks to elaborate on the second point, the sensitivity towards ethically important moments. These moments are difficult to predict, but manifest themselves during the research process for example when the researcher interacts with participants and through the process of gaining knowledge about the lifeworld of participants [7,44]. The following example illustrates the difficulty in identifying and anticipating an ethically delicate moment.

Caroline: "I kind of think it goes here."

(Caroline takes her lasagna photo and puts it with the healthy food)

Clara: "I don't know. I'm not sure, I think this one goes. I don't know."

(Clara removes Caroline's lasagna photo from the healthy food.

Caroline knits her brows, pushes her bottom lip outwards and looks at Clara)

Iben: "Uh uh, Caroline! Killer face."

Clara: "No, but I don't think so. I don't know, I'm sorry."

Iben: "But that stuff that's also healthy."

(Iben points to the lasagna photo)

Caroline: "Yeah, I think so too."

Clara: "It's just that cheese is not like super healthy."

Caroline: "No, but … "

(Focus group with Clara, Caroline, Iben and Filippa)

The quote is from a focus group with four 13-year-old girls, Clara, Caroline, Iben and Filippa. At the time of the focus group, the girls were good friends and described themselves as a "squad". In the excerpt, they are discussing their photo diaries and sorting their pictures into different piles, one of them with "healthy food". As the quote shows, Clara expressed the view that Caroline's photo of lasagna does not belong in the healthy category. After this episode, Caroline withdrew from the conversation and barely said anything for the rest of the focus group. During the recess after the interview, she avoided her friends and hung out with another group of girls. The teacher had not been present during the focus group and was not there during recess to observe Caroline's reaction, and I debated whether I should intervene. On the one hand, I felt it was a bit silly. It was, after all, "just a lasagna", and quarreling with your friends is a part of growing up. On the other hand, I felt that my research had somehow hurt Caroline's feelings. I had seen Caroline crying at school on some occasions, and the teacher had told me "she had issues" and "was a sensitive girl". I thus had the sense that the exercise had reinforced unpleasant feelings and sparked tension between the girls. I knew that it was not just about the lasagna, but a question of friendship, status and identity. In this situation, I chose not to intervene, mainly because I did not know what to do, and I hoped the girls would quickly make up, so the risk of long-term harm would not be great. Instead, I could have followed up with Caroline when I sensed that she was sad after the interview. I could have approached her and inquired into what she was feeling to get more information on how the focus group had influenced her and to assess the potential harm. It is not always possible to follow up with participants, and another strategy I could have chosen was to immediately act on the signs of unease I were sensing and tried to facilitate reflection and reconciliation between the girls during the focus group, which will further be elaborated in Section 4.3. It is difficult to decide on what to do, but I believe a strategy of reflexivity—reflecting critically on the potential problems and responses from the outset of the research—could have helped me respond to this situation in a more suitable way.

A prerequisite for being able to act ethically, is the ability to sense when ethics are at play, and what I want to show with this episode is that it is sometimes difficult for the researcher to anticipate what makes a situation become delicate. I knew that the focus group and the questions could potentially result in uncomfortable conversations, but I had no idea that a discussion on whether lasagna was healthy food could cause so much conflict. The example thus highlights how making sure your research is ethically justifiable is not solely a task you deal with beforehand through procedural ethics, but a process that continues throughout data collection, data management, publication, etc. Doing ethical research thus also involves making decisions about what is appropriate in a specific situation in a specific context, and we need to pay attention to not only procedural ethics, but also to situated and applied ethics [6]. As mentioned, a prerequisite for making appropriate decisions in concrete situations is sensitivity to identifying ethically delicate moments [40]. The question is how this sensitivity can be cultivated. Pader uses the terminology "ethnographic sensibility" [45] to denote awareness of details with orientation towards the meanings of these details in this particular context that characterizes ethnographic research. Ethnographic sensibility thus entails that the researcher sharpens and uses all her senses when doing research. Similarly, I would argue that the sensitivity to identify ethical issues in research can be developed by activating the senses and directing them at identifying ethical issues. This "ethical sensibility" is about awareness of details (for example the tone of Clara's voice, Caroline's facial and bodily expressions) and awareness of the meaning of what is happening in terms of research ethics, i.e., harm to participants. Ethical sensibility thus entails attention to research ethics in the process of conducting research as well as an ability to sense when research ethics come into play.

### 4.2. Constructing and Posing Questions

As scholars, we spend a lot of time on methodological discussions about how to construct questions. Reflections on how to construct and pose questions in order to address ethical issues are a way to minimize the risk of reproducing or reinforcing stereotypes and stereotyped reasoning.

I believe one of the most important parts of asking questions (in interviews and questionnaires) for this purpose is briefing and debriefing to make sure our research lives up to ethical standards. When we formulate briefings and debriefings, we are often concerned with procedural ethics such as informed consent. However, briefings and debriefings can also be used to manage the risk of reproducing and reinforcing stereotypes. For example, in the study I conducted, I started out by establishing some ground rules for the interview by making a statement, such as:

> In this group, it is also very important that you respect what others say, and that you do not repeat it to classmates, teachers, or others afterwards. You can tell what we discussed, but you cannot say that it was X who said it. Do we agree?

Moreover, I finished off the focus group with schoolchildren by asking them questions such as, "How has it been talking about your class and the groups in it?" in order to get a sense of how sensitive it had been for the students to talk about these issues.

Some scholars argue that we should ask open-ended questions instead of statement questions to avoid priming and legitimizing the use of particular stereotypes. As mentioned, whether this is a more ethical strategy is debatable. In my research, I adopted this approach and asked open questions like, "How do you think these photos fit together?", "Which photos or piles of photos do you think best describe the pupils/your peers?" I thus left it to the participants to construct the categories. However, in order not to leave these categories unquestioned, I combined the open-ended questions with making the participants reflect on these categories, which I discuss further in the following section.

Moreover, during interviews with the schoolchildren, I tried to legitimize behavior that was often perceived as "less healthy" or "unhealthy" by using myself as an example:

> You know how some people care a lot about their health and do a lot of things to stay healthy, and others maybe care more about other things? For instance, I don't always think that much about being healthy. I like chocolate a lot, and I really like to watch series on TV, and I sit in front of my computer for hours at work every day.

This meant that if there were children participating in the focus group who belonged to the group of "less healthy" students, there was at least one other person present in the room (me) who also belonged to that group, hopefully making the situation more comfortable.

### 4.3. Facilitating Reflections

Me: "So how do you feel about this? Do you think it is okay or fair that it is this way?"

Karla: "It doesn't have to be different."

Carl: "I think it is fine."

Karla: "I also think it is fine."

Marius: "Yeah."

Mette: "Yeah."

Carl: "I think that maybe sometimes we could hang out with some of the others also."

Karla: "Yes, for example Lise. Sometimes she is left out and that annoys her."

Mette: "When you look at these cards, it kind of makes you think that there are a couple of people that you don't really know like where they belong."

Karla: "Yeah you don't ... "

Mette: "know what to do with them ... "

Karla: "yeah"

Mette: "Because they don't really ... yeah ... "

(Focus group with Karla, Marius, Carl and Mette)

This excerpt is from a focus group with four 13-year-olds (two boys and two girls) who belonged to the group of popular children in the school. The quote is from the part in the interview where they had just categorized themselves and their peers in "friendship groups", showing me with small nametags the hierarchy of the school class. At the end of the interview, I encouraged them to reflect upon their classifications by asking whether they thought it was okay and fair that it was this way. As the quote shows, they started out by agreeing that it was fine, but during the conversation, they actually discussed the problems in the class. This illustrates that by making participants explain and discuss their reasoning (for example in a focus group), it is possible that the research situation can challenge existing hierarchies, which becomes clear in the statement made by the girl, Mette, "when you look at these cards". Doing the exercises and answering the questions, which potentially reproduce and reinforce stereotypes and stigmas, allow reflection and challenging of stereotyped reasoning and stigmatization. Another strategy for researchers could thus be to facilitate reflections; encourage participants to reflect critically upon their answers.

## 5. Conclusions

This paper discusses how ethical challenges can arise when we study categorization and stigmatization. More specifically, how we can avoid reinforcing such processes while still conducting efficient research. This problem, I argue, is seldom addressed comprehensively in the literature. While it may not be possible to eliminate the risk of reinforcing these processes, I argue that there are steps one could take to try to minimize it. First, scholars should use reflexivity not just as a strategy to ensure the quality of the knowledge claims that they make, but also in relation to ethical issues. This includes developing an ethical sensibility. A pre-requisite for making ethically appropriate choices is that researchers pay attention to and are able to identify ethically delicate moments, which may not always be straightforward. How the research process plays out—especially in qualitative research—is not easy to anticipate and what can turn out to be an ethically delicate moment in a particular research context is thus to a large extent unpredictable. Thus, the researcher should attune her senses to the ethical. Furthermore, the researcher should not only think about methods, but also ethics when constructing and posing questions, for example by using briefings and debriefings to not only secure procedural ethics, but also to inquire into how the participants experience the interview and thereby facilitating the researcher's critical reflections on his or her research practice. A final strategy is to make participants reflect upon their opinions and answers. After all, it is possible that the research situation will actually challenge stereotyped reasoning among participants. Ethical reflexivity and ethical sensibility are important for conducting ethical research. While this article has focused on the individual researcher, I would argue that a crucial part of developing self-reflexive strategies is deliberation within the research community about research ethics.

**Funding:** This research received no external funding.

**Acknowledgments:** I would like to thank the reviewers for excellent comments on previous versions of this article as well as Ana Patrícia Hilário and Fábio Rafael Augusto for constructive feedback and the possibility to contribute to this special issue of Societies. Finally, I would like to thank Annette Bruun Andersen for providing language editing.

**Conflicts of Interest:** No conflict of interest to declare.

# Appendix A

**Table A1.** Interview guide 1: Social categories and identities in the classroom.

| Phenomenon | Research Questions | Questions |
|---|---|---|
| Briefing | Presentation and warm-up questions | (1) To begin with, I would like you to state your name and your age, so that it will be on the tape<br>Before we begin the interview, I would just like to say that whatever you say here is confidential. This means that I will not repeat what you say to your teacher, to the other children from your class, or to your parents. I will use it in my paper, but no one will know that you were the ones who said it. In this group, it is also very important that you respect what others say, and that you do not repeat it to classmates, teachers, or others afterwards. You can tell what we discussed, but you cannot say that it was X who said it. Do we agree?<br>(2) Can you tell me a bit about how it is like in your class? |

*Introduction to activity based questions: "pile sorting"/"card-sorting-task" (joint task)*
*A pile of name tags with all the names the pupils from class is spread across the table*

| Phenomenon | Research questions | Questions |
|---|---|---|
| Groups in the class<br>Mapping of the social landscape<br>(Group level)<br>Underlying categories and principles of differentiations | Who belongs with whom?<br>Who does not belong with whom?<br>What do the pupils in the different groups have in common? How do they resemble each other?<br>Are there some points where they differentiate? Which?<br>How do the pupils in the different groups separate themselves from each other? Are there some points where they resemble each other?<br>How are the groups' mutual relations? | Here are a bunch of name tags with the names of the pupils in your class. Sometimes there are some who hang out more often in a class. Well, some groups of people who hang out and talk more than they do with others. That happens at my work too. For example, I talk a lot with a guy called Jonas because we share an office. That does not mean that you do not like other people. There might just be somebody you hang out with more often.<br>(2) How is it like in your class? If you had to group the people in your class together the way they belong, how would you do so?<br>You can decide how big the groups should be. They do not need to be of equal size, and some can be alone. That is up to you. There is not a correct way or a wrong way of doing it. You are the ones who decide.<br>(3) What do the pupils in the groups have in common?<br>(3b) How are they different from each other?<br>(3d) Why do you think that these pupils *hang out*?<br>(3e) It is always like this? Are there different groups during class, during recess, or outside school?<br>(3f) How are the pupils in the various groups different from each other?<br>(3g) Are there some points where they resemble each other? (agreement?)<br>(4) How do you think it would look like if your teacher had made the groups? |
| Self-identification (disidentification)<br>(individual level) | Which group of pupils does the individual pupil associate himself/herself with?<br>Which pupils do they distance themselves from? Why?<br>Which groups do they associate/distance themselves with/from? | I can see that you have placed yourselves there.<br>(5) Which groups do you think you fit minimally into?<br>(5a) For example, if you were going camping, and the teacher decided whose tent you slept in/which cooking team you were on, and you ended up in group X. How would you feel about that? |
| Debriefing | | It is completely normal that you sometimes hang out with some people in class more so than with others. That does not mean that you do not like other people.<br>(6) How has it been talking about your class and the groups in it?<br>(6a) Do you sometimes talk about it in class? For example, during form time?<br>(6b) Do you sometimes talk about it with your classmates? For example, in your spare time? |

**Table A2.** Interview guide 2: Health categories and identities in the classroom.

| Phenomenon | Research Questions | Questions |
|---|---|---|
| Introduction | Presentation of the participants in the focus group | (1) Like last time, we are going to start the round with you stating your names, so it will be recorded on the tape |
| **Activity-based questions: Photo diaries (individual task)** | | |
| Sense of health and categories (individual) | What is healthy? What is unhealthy? According to the pupils, when are they being healthy/unhealthy? How does health appear in their day-to-day life? | You have sent me some pictures that show healthy and unhealthy things, activities, times, etc. in your everyday life. (2) Would you mind giving a brief account of your pictures? Let us take a round … (2a) On picture X, is that something you often do/experiences/eat in your everyday life? (2b) Do you think what is on picture X is healthy/unhealthy? Why? (2c) When does it happen? (2d) Who are you with when it happens? Where are you? (2e) Do you think about if it is healthy/unhealthy when you do/eat what is on picture X? Why/Why not? (2f) Do you talk about how healthy/unhealthy it is when you do it? |
| **Activity-based questions: joint task—"Picture sorting" with health pictures(Joint task)** | | |
| Health categories (joint) | What is healthy/unhealthy? Which differences and similarities of the perception of what is healthy and unhealthy are there between the pupils? How is that reflected? | (3) We have looked at your pictures. If you had to group your pictures the way they fit together, how would you do it? Please go ahead. There does not necessarily need to be a pile with what is healthy and another one with what is unhealthy. You can make multiple piles and think about it. *Probes* *(3a) All right, can you tell me something about the piles you have made?* *(3b) Why have you chosen to divide them into these piles?* *(3c) Do you all agree, or are there some of the pictures or the piles you disagree with? How would you like them to look?* |
| **Activity-based questions: "Picture sorting" with health pictures and name tags** | | |
| Health identities and categories | How do they understand the other pupils' health? Who is healthy/unhealthy? Who is like each other in regard to their health? Why? How do the pupils understand their own health? With what do the pupils associate themselves? From what do they distance themselves? With whom do they associate themselves? From whom do they distance themselves? | I have these name tags from last time. Now, it is normal that some people care deeply about their health and want to do a lot to stay healthy while others care about other things. For example, I do not always care if I am healthy (I really like chocolate, and I also like watching shows on the television). I would probably place myself there. *(they get their own name tag)* (4) If you had to place yourselves in one of the piles with the pictures, where would that be? Why? (4a) If you had to place the others from that class, how would you do it? Why? *(agreement?)* |
| Health promotion | | (5) Do you sometimes talk about what is healthy and unhealthy with each other? (5a) When for example? (5b) Do you listen to what your classmates say about health? (5c) For example, if one of your classmates began to eat healthier or started exercising, would you do the same? (6) Do you also talk about health with your teachers at school? (6a) Is health something you learn about at school? Something that is in the curriculum? (6b) What do you talk about then? (6c) What do you do? (6c) Do you like it when you learn about health, or when you move around? (6d) Do you listen to what your teachers (or the health visitors) say about health? (7) Do you sometimes talk about what is healthy or unhealthy with your parents? (7a) What do your parents say? (7b) How you respond? (7c) Do you listen to what your parents say about health? (8) Who do you listen to the most? |

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
