# Peer review of "Reinforcing and Reproducing Stereotypes? Ethical Considerations When Doing Research on Stereotypes and Stereotyped Reasoning"

_societies, doi:10.3390/soc9040079_

Round 1

Reviewer 1 Report

The literature review could be expanded to include stigmatization literature in general, ie what are the main ways this is studied? What are some example questions in both qualitative and quantitative research that the author would point to as examples of being problematic and concerning in light of the ethical concerns raised? Does/has anyone explored how measuring stigmatization reinforces and/or reproduces those stereotypes (ie maybe through priming?) This would help situate the ethical concerns that are raised within the broader literature on this topic.   Returning to this in the conclusion and providing specific instances of when the three steps could be utilized to improve this type of research would provide a much stronger contribution to the literature. Illustrating how these steps would help this field, not by calling out any one article in particular, but how researchers could improve and enact these steps would benefit the paper.

In the study by Gilliam, was evidence provided to support that the 'questioning of the boundaries' was effective?

In the example with Caroline, did the author follow up in any way?  What would the author suggest should or could have been done in this situation?   

In the results/conclusion, could example questions that 'don't encourage and legitimize the use of racial or class stereotypes' be provided? 

Author Response

First, I would like to thank you for your helpful suggestions, which I am sure will improve the article.

Regarding the literature review, I have tried to expand this by including a presentation of how stereotypes are usually studied in various literatures and within various methodological approaches.

“In the study by Gilliam, was evidence provided to support that the 'questioning of the boundaries' was effective?” I have addressed this in line 111-113.

“In the example with Caroline, did the author follow up in any way?  What would the author suggest should or could have been done in this situation? “ I have tried to address this comment in line 264-284.  

“In the results/conclusion, could example questions that 'don't encourage and legitimize the use of racial or class stereotypes' be provided?” I have specified how questions could be asked in section 4.2 as well as provided one example in the conclusion.

Kind regards

Reviewer 2 Report

This paper by an educational researcher grapples with the difficulty of potentially reinforcing stereotypes or causing harm to children as research participants (and to a lesser extent, with adult participants). I thought it was an interesting topic, but a blind spot that was immediately apparent in the abstract was the lack of a discussion of research ethics approval. Human research ethics committees or institutional review boards in most jurisdictions have the explicit role of evaluating potential risks to human participants in research (although sometimes extremely low risk research is excluded from requiring review). I have a few concerns and queries, with suggestions for places where the discussion of this work might be expanded, which I hope the author will find helpful.

- I was surprised that the author only briefly raised this point as an aside (line 239-239). I think it would therefore be useful to expand a bit more on this issue. Are ethics review boards effective in weighing risks to participants, considering issues of informed consent and potential benefits from research? Or is it a mere bureaucratic step that does not actually improve the ethical justifiability of a particular research study?

- On a related note, was the author's own research approved by an ethics board? I would have assumed that this would be required given that s/he is working with young children. Was informed consent obtained from parents and teachers on behalf of the participants in her research?

- Similarly, the researcher discusses the conflict between Clara and Caroline in purely negative terms. But these are 13 year old children and the conflict was observed  by the researcher. Was this incident related to a teacher who could provide an appropriate pastoral intervention?

- Moreover, since the relationships are pre-existing, it is difficult to draw any strong conclusions from this anecdote. Was this, on balance, a negative experience for Caroline? For example, within their "squad" was there always tension between Clara and Caroline? Were the other girls more welcoming of Caroline? Is dealing with conflict and disagreement a normal part of growing up that doesn't really pose much long term risk even if participants argue a bit in a focus group?

- Regarding the issue of whether asking participants to agree or disagree with a statement might reinforce that particular stereotype, I was not convinced that the survey question approach was that bad, nor was I convinced that the suggested alternative was an improvement. In fact, I suspect they might be measurements of different things. Asking someone whether they agree or disagree with a statement relies very heavily on their conscious or explicit attitudes. I think the author needs to present or cite some empirical evidence that these kinds of explicit questions cause a priming effect that increases racial stereotyping (the literature used in support of this argument is fairly weak, based on a single book). On the other hand, asking participants to sort pictures into categories might access more implicit attitudes and beliefs - perhaps without the participants realising it (this approach reminded me of the famous Harvard Implicit Association Test). In this case, an argument could be made that activating the implicit memories and then leaving them unchallenged could be even worse than activating a conscious attitude and asking the person to explicitly declare whether they agree or disagree with a statement.

Author Response

Thank you for these constructive comments and helpful suggestions.

Regarding the point about the lack of discussion of ethical review board, this is a very relevant critique, which I have tried to address in the following lines:

Line 41-51, 212-217, 234-237, 283-291

Although the article focus on ethics in practice the role of ethical review board should of course be mentioned and discussed, which I hope I have succeeded in with the revisions.

Regarding the point about the conflict between Clara and Caroline, I have tried to describe more in detail and nuance the situation in order to make my point more explicit (line 264-282)

“Regarding the issue of whether asking participants to agree or disagree with a statement might reinforce that particular stereotype, I was not convinced that the survey question approach was that bad, nor was I convinced that the suggested alternative was an improvement. In fact, I suspect they might be measurements of different things. Asking someone whether they agree or disagree with a statement relies very heavily on their conscious or explicit attitudes. I think the author needs to present or cite some empirical evidence that these kinds of explicit questions cause a priming effect that increases racial stereotyping (the literature used in support of this argument is fairly weak, based on a single book). On the other hand, asking participants to sort pictures into categories might access more implicit attitudes and beliefs - perhaps without the participants realising it (this approach reminded me of the famous Harvard Implicit Association Test). In this case, an argument could be made that activating the implicit memories and then leaving them unchallenged could be even worse than activating a conscious attitude and asking the person to explicitly declare whether they agree or disagree with a statement.”

Thank you very much for this comment. I do agree, and I have changed the section 4.2. and tried to incorporate this point and nuance the debate.

Round 2

Reviewer 1 Report

The author has addressed the previously noted concerns.  However, the additional information about how the author handled the situation that arose during the focus group with the participants was overall disappointing and left me wondering what do we do in these situations?  I feel the author has highlighted an important and serious concern but the solution offered is vague and ultimately leaves me wondering "So what do we do with this?" 

I appreciate the author's honesty when they write, "mainly because I did not know what to do, and I hoped the 278 girls would quickly make up, and the risk of long-term harm would not be great. However, I could 279 have followed up with Caroline when I sensed that she was sad after the interview. I could have 280 approached her and inquired into what she was feeling to get more information on how the focus 281 group had influenced her and to assess the potential harm."  However, it may not be possible in all situations to address or follow up with the participants.  How can researchers do something in the moment and address these concerns as soon as they are raised? 

Author Response

Thank you very much for the comment which I completely agree with. I have tried to elaborate on what could be done (line 290-300). This discussion could perhaps be also be unfolded even further, but I have to be honest (again) and admit that I would need a bit more time for that. 

Thank you once again for your suggestions

Kind regard

Reviewer 2 Report

I appreciate the author taking the time to revise their manuscript. However, the lack of markup in the PDF made it difficult to see what parts of the text were revised.

The author claims that Danish universities have no ethics committee procedures. I was surprised by this and tried to verify it, but not being familiar with the Danish system I am sure I didn't get a complete picture. I recommend that the author give a few sentences or a paragraph to contextualise the Danish system - it appears to me that there are ethics committees, but that these are not localised to universities and only apply to health/medical research. Is this correct? It may also be worth mentioning that the Danish system does not have ethics committees for this type of research in the abstract - many researchers in other countries are accustomed to this requirement.

On a similar note, does the author think that introducing such a system for social/behavioural research in Denmark would be beneficial? Why or why not?

There was also a second point I was making about research approaches that was unaddressed in section 4.2. In changing your approach to reduce the risk of reinforcing stereotypes, you may also change what you are measuring.

Author Response

Thank you very much for the comments. 

"I appreciate the author taking the time to revise their manuscript. However, the lack of markup in the PDF made it difficult to see what parts of the text were revised."

I am really sorry. This time, I have highlighted the changes with yellow.

"The author claims that Danish universities have no ethics committee procedures. I was surprised by this and tried to verify it, but not being familiar with the Danish system I am sure I didn't get a complete picture. I recommend that the author give a few sentences or a paragraph to contextualise the Danish system - it appears to me that there are ethics committees, but that these are not localised to universities and only apply to health/medical research. Is this correct? It may also be worth mentioning that the Danish system does not have ethics committees for this type of research in the abstract - many researchers in other countries are accustomed to this requirement."

I have tried to provide some insights into the Danish system: Line 216-224. I have not mentioned this in the abstract, as I could not quite see where this information would fit.

"On a similar note, does the author think that introducing such a system for social/behavioural research in Denmark would be beneficial? Why or why not?"

I would rather not comment explicitly on the Danish case as I would rather wait and see how the introduction of these “not mandatory” IRBs function, but I have commented on the role of ethical review boards in general line 43-52.

"There was also a second point I was making about research approaches that was unaddressed in section 4.2. In changing your approach to reduce the risk of reinforcing stereotypes, you may also change what you are measuring"

Thank you for this comment. I have chosen to address this in another section as I thought it was appropriate to mention in that context line 117-125.

kind regards